# Revised Model of Abrasive Water Jet Cutting for Industrial Use

**DOI:** 10.3390/ma14144032

**Published:** 2021-07-19

**Authors:** Libor M. Hlaváč

**Affiliations:** Department of Physics, VSB–Technical University of Ostrava, 17. listopadu 15/2172, 708 00 Ostrava, Czech Republic; libor.hlavac@vsb.cz; Tel.: +420-59-732-3102

**Keywords:** abrasive water jet, modelling, cutting, process control, industrial application

## Abstract

Research performed by the author in the last decade led him to a revision of his older analytical models used for a description and evaluation of abrasive water jet (AWJ) cutting. The review has shown that the power of 1.5 selected for the traverse speed thirty years ago was influenced by the precision of measuring devices. Therefore, the correlation of results calculated from a theoretical model with the results of experiments performed then led to an increasing of the traverse speed exponent above the value derived from the theoretical base. Contemporary measurements, with more precise devices, show that the power suitable for the traverse speed is essentially the same as the value derived in the theoretical description, i.e., it is equal to “one”. Simultaneously, the replacement of the diameter of the water nozzle (orifice) by the focusing (abrasive) tube diameter in the respective equations has been discussed, because this factor is very important for the AWJ machining. Some applications of the revised model are presented and discussed, particularly the reduced forms for a quick recalculation of the changed conditions. The correlation seems to be very good for the results calculated from the present model and those determined from experiments. The improved model shows potential to be a significant tool for preparation of the control software with higher precision in determination of results and higher calculation speed.

## 1. Introduction

The period of extensive use of water jet and abrasive water jet industrial application started more than 50 years ago. The first models for water jets were presented in the 1970s [1,2]. During the 1980s, Hashish presented his studies of abrasive water jets [3,4]. More recently, in the beginning of the 1990s, Hlaváč presented his theoretical model prepared for both water and abrasive water jets [5]. Later on, during the 1990s and in the beginning of the 21st century, many researchers all over the world continued modelling activities aimed at abrasive water jet machining. Regression-based models were prepared [6] and the first models for 3D machining were published [7,8]. Models aimed at including inherent phenomena have been prepared, namely, those describing processes in the mixing chamber and the focusing tube [9], in the interaction area [10,11] and some geometrical phenomena of the cutting processes. The research teams focused on processes with substantial impact on product geometry: origin of the cutting surface [12,13,14], formation of the striations [15,16,17] and the taper [18,19], determination of the jet lag inside kerf [20] and influence of the cutting parameters on the kerf geometry [21,22].

In recent years, many specific research applications have grown in importance. First of all, there are investigations aimed at some specific materials like thick metals [23] and titanium alloys for aerospace [24,25]. Some other interesting materials are glass [26] or composites [27]. Very interesting research was performed by Akkurt [28], who studied the front characteristics in applications of the AWJ on brass. Some recent investigations focused on a suppression of the negative impact of the trailback and the taper on the product quality [29], namely, in machining of composites [30,31]. Other researchers focus on various mathematical—numerical methods and their applications in AWJ processes. These methods are, namely, finite element methods (FEM) [32,33], smoothed particle hydrodynamics (SPH) [34] and computational fluid dynamics (CFD) [35].

The investigation of AWJ machining cannot be adequate without research of abrasive materials. Some researchers focused on investigations of the disintegration of abrasive particles in the mixing process [36], others searched for new types of abrasive materials [37], others studied the recycled ones [38,39] or try to apply ice particles [40], as the most environmentally friendly abrasive. Some special applications like using the AWJ for sculpturing [41], micro-piercing [42] or micro-machining [43] bring new challenges for researchers dealing with water jetting. However, the studies of applications of AWJ on the “classical” materials like tiles [44], wood [45] or Hardox [46] can also bring about new knowledge. The problem of surface quality is often closely tied in with the surface roughness. Therefore, many investigations were performed to assess, predict or monitor this parameter [47,48,49,50]. A few publications were also addressed at the reliability and safety problems of water jet technologies [51] or other related problems. However, preparation of the theoretical and more complex models is quite rare in the last decade. Therefore, partial models focused on either the mixing processes [52], modelling of the abrasive particles energy [53], or on the preparation of some superstructures for previous models [54,55], or in combination, have been presented. Recent research effort is focused on numerical modelling of abrasive water jets, e.g., the impact on material and the respective kerf shapes [56] or modelling of the micro-machining process imprint shapes [57]. However, the upgraded analytical model of the abrasive water jet that is presented in this article can be a new direction for the innovation of control systems for abrasive water jetting facilities and the development of some new devices, improved software systems, or in combination.

## 2. Plain Waterjet—Material Interaction: Summary of the Initial Physical Model

The model of water jet (WJ) interaction with material has been derived by applying the analytical physical description of the interaction process and it was presented in [58]. The derivation of this initial model was based on application of basic physical principles—the energy conservation law and the momentum conservation law. The resulting two equations are necessary for a development of the subsequent abrasive water jet model. Therefore, the most important forms presented in [58] are summarized here:(1)$$h_{n+1}=\frac{\pi \;d_{o}\;\sqrt{2\rho_{o}\mu^{3}\gamma_{p}^{3}p_{o}^{3}e^{-5(\xi L\;+\;\xi^{*}h_{n}^{*})}}\;(1\;-\;\alpha_{n}^{2})\;\cos\;\theta }{8\;\chi \;\rho_{M}\;{\tilde{v}}_{P}^{\frac{\rho_{o}}{\rho_{M}}}\;\left(\alpha_{n}^{2}\;e^{-2(\xi L\;+\;\xi^{*}h_{n}^{*})}\mu \;\gamma_{p}p_{o}\;+\;\frac{\rho_{o}}{\rho_{M}}\;\sigma \right)}$$
(2)$$\alpha_{n}\;=\;1\;-\;\frac{C_{f}^{2}\;\sqrt{2\mu^{3}\gamma_{p}^{3}p_{o}^{3}}\;\rho_{M}^{*}\;k^{*}}{8\;\sqrt{\rho_{o}}\;\eta \;\sigma_{s}\;a\;e^{3(\xi L\;+\;\xi^{*}h_{n}^{*})}}$$
(3)$$h_{n}^{*}\;=\;h_{1}\;+\;h_{2}\;+\;h_{3}\;+\;\ldots \;+\;h_{n-1}\;+\;h_{n}$$

The first equation describes the depths of penetration of the pure water jet into a material in the (*n* + 1)-th pass along the same trace. The axis of the impinging jet is tilted from the perpendicular to the surface in the “plane of cut” for the angle θ. The “plane of cut” is determined by the traverse speed vector and the jet axis. The second equation calculates the energy absorption coefficient α after the n-th pass of the jet along the same trace. The third equation summarizes the penetrations of the individual passes to the total depth of penetration. The equations have been used in the expression for one pass and without tilting for the basic preparation of the AWJ model through the principles of similarity. Therefore, the AWJ derivation starts from the application of these two equations for calculations of the pure water jet’s impact on materials:(4)$$h\;=\;\frac{\pi \;d_{o}\;\sqrt{2\rho_{o}\mu^{3}\gamma_{p}^{3}p_{o}^{3}e^{-5\xi L}}\;(1\;-\;\alpha^{2})}{8\;\chi \;\rho_{M}\;{\tilde{v}}_{P}^{\frac{\rho_{o}}{\rho_{M}}}\;\left(\alpha^{2}\;e^{-2\xi L}\mu \;\gamma_{p}p_{o}\;+\;\frac{\rho_{o}}{\rho_{M}}\;\sigma \right)}$$
(5)$$\alpha \;=\;1\;-\;\frac{C_{f}^{2}\;\sqrt{2\mu^{3}\gamma_{p}^{3}p_{o}^{3}}\;\rho_{M}^{*}\;k^{*}}{8\;\sqrt{\rho_{o}}\;\eta \;\sigma_{s}\;a\;e^{3\xi L}}$$

However, several additional specific phenomena typical for AWJ must be introduced into the model to prepare fairly reliable equations, as is described in the next sections.

## 3. Differentness in Description of the Abrasive Water Jet Regarding Pure Water Jet

The most important difference between the AWJ and the pure water jet is the presence of solid-state abrasive particles inside the resulting jet (flow). This contribution is aimed at the injection AWJ, but some resulting formulas can be applied even for the slurry jets. Nevertheless, the process of abrasive mixing with the water jet, specific for the injection jets, needs deeper analysis, because, contrary to the slurry jets, abrasive particles undergo a substantial size change during the mixing process. The abrasive particle size change has been studied many times for various purposes in the past, e.g., as a tool for the intended disintegration of some minerals or coal [59,60]. These studies have shown a substantial reduction of the particle size in the mixing process. Therefore, the equation for calculation of the abrasive particle size change needs to be included into the system of equations applied in the control loop of the cutting system. It can be used in this form (see [59]):(6)$$a_{n}\;=\;\frac{a_{o}}{1+\frac{C_{D}\pi d_{o}^{2}\mu_{o}^{2}p_{o}^{2}\gamma_{o}^{2}}{24\rho E_{P}a_{o}c_{o}c}}$$

The next specific parameters modifying efficiency of the AWJ are connected with the energy losses and the momentum changes inside the mixing chamber and the focusing tube. To define these phenomena more precisely, Hlaváč introduced four coefficients modifying velocity of the mixture [9]. The first one modifies velocity according to the suction capacity when the number of incoming particles exceeds the limit for the proper mixing and particles acceleration ($C_{1}$):(7)$$C_{1}\;=\;\frac{\pi \;\rho_{a}\;v_{i}\;a_{o}^{3}}{3\;d_{o}\;q_{a}}\;\mathrm{For}\;\frac{3\;q_{a}}{\pi \;\rho_{a}\;a_{o}^{3}}\;\leq \;\frac{v_{i}}{d_{o}}\;\Rightarrow \;C_{1}\;=\;1$$

The second coefficient ($C_{2}$) indicates smooth flow, as the number of disintegrated particles passing through the focusing tube cross-section is limited:(8)$$C_{2}\;=\;\frac{d_{a}}{\sqrt{6}\;a_{n}}\;\mathrm{For}\;d_{a}\;>\;\sqrt{6}\;a_{n}\;\Rightarrow \;C_{2}\;=\;1$$

Modification of the velocity due to the friction inside the focusing tube with respect to the material of the tube indicates coefficient $C_{3}$:(9)$$C_{3}\;=\;\left(1\;-\;f\;l_{a}\right)$$

The fourth coefficient ($C_{4}$) modifies the mixture velocity according to the size relations in the system “orifice diameter—abrasive particle size—focusing tube diameter”. Considering the non-zero number of the original sized particles in the mixture and their smooth flowing through the system, the coefficient $C_{4}$ can be determined as:(10)$$C_{4}\;=\;1\;-\;\frac{{\left(a_{o}+a_{n}+d_{o}\right)}^{2}}{d_{a}^{2}}\;\mathrm{for}\;d_{a}\;>\;0$$

Equations (7)–(10) are presented here for better understanding of the final model preparation. The equations determine the coefficients applied for calculation of the resulting mixture velocity from the momentum conservation law.
(11)$$v_{a}=C_{1}C_{2}C_{3}C_{4}v_{i}\frac{q_{w}}{q_{w}+q_{a}}$$

It is evident that Equation (11) is similar to the one already presented in the beginning of the modelling attempts [61]. It is because it is derived in the same manner, from the momentum conservation law. The coefficient of momentum transfer efficiency η used by Hashish in [61] is replaced here by the product of four coefficients being calculated or set by the logical mathematical operations according to the actual state of the cutting process variables and the mixing conditions.

## 4. Model of the AWJ Transformed from the WJ Model

The AWJ model prepared by a direct transformation from the pure water jet model is based on analogy presumption. It is supposed that by transforming the liquid density and pressure and using the appropriate quantities in Equations (4) and (5) it is possible to obtain the equation for the depth of the AWJ penetration into material [9]. The first step is to determine the density of the mixed flow (water with abrasive)—this is determined by Equation (12).
(12)$$\rho_{j}=\frac{4\rho_{a}(q_{w}+q_{a})}{\pi \rho_{a}v_{i}d_{o}^{2}+4q_{a}}$$

The next step is to determine the respective pressure of the mixed flow, i.e., liquid with the density determined using Equation (12). This pressure is calculated from Equation (13).
(13)$$p_{j}=\frac{1}{2}\rho_{j}v_{a}^{2}$$

Once these most important quantities are determined, the subsequent substitution is applied in Equations (4) and (5): $\rho_{o}\to \rho_{j}$, $\mu \gamma_{p}p_{o}\to p_{j}$, $\xi \to \xi_{j}$, $\alpha \to \alpha_{e}$. Other variables, especially the material properties like the density, the grain size, the strength (both the compressive/tensile and the shear) remain identical for the AWJ as for the pure water jet. The only specific characteristic is the permeability (used in the case of the pure water jet), because it is necessary to select some appropriate property instead for the AWJ cutting, particularly for steels and other metals, carbons, carbides and other materials with low water absorption and penetration. The material hardness *K* (or *HV*) was selected as the proper variable for the AWJ machining and the respective equations were transformed into the forms presented in Equations (14) and (15), see also the citation [14], or Equation (16), see [55]. The exponent used for the traverse speed (number 1.5 instead of the ratio of the jet and the cut material densities) has been determined from the regressions of a huge amount of experimental data obtained on many cut materials in the late 1980s and during the 1990s.
(14)$$h_{lim}\;=\;\frac{C_{A}\;S_{p}\;\pi \;d_{o}\;\sqrt{2\rho_{j}p_{j}^{3}\;e^{-5\xi_{j}L}}\;(1\;-\;\alpha_{e}^{2})}{8\;{\left(v_{P}+v_{Pmin}\right)}^{1.5}\;\left(\rho_{m}p_{j}\alpha_{e}^{2}\;e^{-2\xi_{j}L}\;+\;\rho_{j}\sigma_{m}\right)}$$
(15)$$\alpha_{e}=1-\frac{\sqrt{{2p}_{j}^{3}}\;K\;t_{i}}{8\sqrt{\rho_{j}}\;\sigma_{m}a_{m}}$$
(16)$$v_{Plim}={\left[\frac{C_{A}S_{P}\pi d_{0}\sqrt{2\rho_{j}p_{j}^{3}e^{-5\xi_{j}L}}\left(1-\alpha_{e}^{2}\right)}{8H\left(\rho_{m}p_{j}\alpha_{e}^{2}e^{-2\xi_{j}L}+\rho_{j}\sigma \right)}\right]}^{\frac{2}{3}}-v_{Pmin}$$

Nevertheless, problems with determination of the proper interaction time $t_{i}$ in the efficiency coefficient $\alpha_{e}$ during the real time calculations in new control software being prepared in the last few years lead to the repeated detailed analysis of the process. This analysis uncovered the very important detail. During preparation of the original model of the AWJ [9] the above-mentioned substitutions were input. Subsequently, the correlation between experimental data and results calculated from that theory led to the introduction of the power 1.5 for the traverse speed. All further works were influenced by these decisions. Nevertheless, the experiments used for correlation were influenced by lower efficiency of the system generating the AWJ accompanied by the low accuracy of the devices measuring the pressure inside the high-pressure part of the system. The results were also influenced by less information about the abrasive material and its disintegration during the mixing process. All analyses, performed in the last few years, show that exponent applied at the traverse speed (1.5) is too high and the proper exponent is much closer to the value resulting from the original theoretical physical derivations, i.e., to “one”. Therefore, the exponent has been decreased from the value 1.5 to the value “one” (derived originally from the basic equations for the energy and the momentum conservation laws). Another possible substitution related to the change from pure water to the abrasive mixture, $d_{o}\to d_{a}$, needs to be largely discussed. Although it seems logical that the focusing tube diameter should be used instead of the nozzle/orifice diameter, this substitution needs to be evaluated from the physical point of view. Provided that the diameter of the nozzle represents the amount of energy delivered to the interaction area, it cannot be replaced by the focusing tube diameter, because this change does not increase the energy portion (better said it reduces it). Nevertheless, if the diameter of the nozzle represents the amount of destroyed material, it could be replaced by the focusing tube diameter. Therefore, the actual equations for calculation of the jet penetration limit into the material, Equation (17), or the traverse speed limit for cutting of the selected thickness of material, Equation (18), have the subsequent forms:(17)$$h_{lim}\;=\;\frac{C_{A}\;S_{p}\;\pi \;d_{a}\;\sqrt{2\rho_{j}p_{j}^{3}\;e^{-5\xi_{j}L}}\;(1\;-\;\alpha_{e}^{2})}{8\;\left(v_{P}+v_{Pmin}\right)\;\left(\rho_{m}p_{j}\alpha_{e}^{2}\;e^{-2\xi_{j}L}\;+\;\rho_{j}\sigma_{m}\right)}$$
(18)$$v_{Plim}=\frac{C_{A}S_{P}\pi d_{a}\sqrt{2\rho_{j}p_{j}^{3}e^{-5\xi_{j}L}}\left(1-\alpha_{e}^{2}\right)}{8H\left(\rho_{m}p_{j}\alpha_{e}^{2}e^{-2\xi_{j}L}+\rho_{j}\sigma \right)}-v_{Pmin}$$

Supplementary equation for calculation of the efficiency coefficient $\alpha_{e}$ from the cutting process parameters, the material properties and other factors remains unchanged, commonly expressible in this form:(19)$$\alpha_{e}=1-\frac{\sqrt{2p_{j}^{3}}Kt_{i}}{8\sqrt{\rho_{j}}a_{m}\sigma_{s}}$$

Equation (17) or Equation (18), together with Equation (19) and, of course, with the preceding ones, (6)–(13), are directly applicable in practice, both for predictive or analytical calculations. They can be also used for a preparation of the operating and control software for machining systems with AWJ (as was proved in the past with their older forms). Calculation of the coefficient $\alpha_{e}$ is much more convenient in practice from evaluation of the experimental cut made in material, especially when machining of materials with the untrustworthy material information. The appropriate relation for calculation of the coefficient $\alpha_{e}$ is then expressed as Equation (20):(20)$$\alpha_{e}\;=\;\sqrt{\frac{C_{A}S_{p}\pi d_{a}\;\sqrt{2\rho_{j}p_{j}^{3}\;e^{-5\xi_{j}L}}\;-8h\left(v_{P}+v_{Pmin}\right)\rho_{j}\sigma_{m}}{C_{A}S_{p}\pi d_{a}\;\sqrt{2\rho_{j}p_{j}^{3}\;e^{-5\xi_{j}L}}\;+8h\left(v_{P}+v_{Pmin}\right)\rho_{m}p_{j}e^{-2\xi_{j}L}}}$$

In the case of machining processes other than cutting, additional equations may be necessary, describing some specific factors and the specific behavior of the jet in the respective processes. It is also important to add some equations describing the compensation of the jet delay and the taper during the cutting process [19,55]. The modification of the traverse speed, for meeting the requirements of a certain surface quality, can be calculated from Equation (21), introduced, e.g., in [55] and completed by $C_{Q}$ relation to the cut material behavior and the AWJ abrasive material type; the condition $C_{Q}\in (0\;;\;1\rangle$ needs to be fulfilled, because the traverse speed needs to fulfil condition $v_{P}\in (0\;;\;v_{Pmin}\rangle$.
(21)$$v_{PQ}=C_{Q}v_{Plim}$$

Similarly, an analogous equation can be derived for a determination of the depth in material with a selected quality of the side walls securable at the set traverse speed. However, the proper relation between the wall quality parameters and the respective traverse speed and other machining parameters needs further investigation for each material and its thickness. The values $h_{lim}$ or $v_{Plim}$ are determined from the actual $v_{P}$ and h applying Equation (22) or Equation (23).
(22)$$h_{lim}=h{\left(\frac{\vartheta_{lim}}{\vartheta }\right)}^{\frac{2}{3}}$$
(23)$$v_{Plim}=v_{P}{\left(\frac{\vartheta_{lim}}{\vartheta }\right)}^{\frac{2}{3}}$$

Respective declination angles determining the cutting wall quality are calculated from Equation (24) or Equation (25) within the scope of the jet energy proportionate to the material thickness.
(24)$$\vartheta =\vartheta_{lim}{\left(\frac{h}{h_{lim}}\right)}^{1.5}$$
(25)$$\vartheta =\vartheta_{lim}{\left(\frac{v_{P}}{v_{lim}}\right)}^{1.5}$$

The usual angle limit for materials with the thickness correlating with the jet energy has a value of 45°, because both the cutting and the deformation wear of material are present during the jet–material interaction. However, if the material thickness exceeds the jet energy capacity, the deformation removal of material (the deformation wear) is impossible. The jet reflects from the kerf bottom in such a case. Therefore, the angle limit corresponds to the maximum angle for the cutting mode, i.e., approximately 22.5°. Very thick material pieces can also be cut, but it is necessary to set the traverse speed so low that the uncertainties in the cutting conditions (pressure, abrasive mass flow rate, material properties local change) cannot induce conditions for deformation wear (then the jet immediately starts to reflect from the kerf). Therefore, the declination angle limit further decreases to only 15° or less. This limitation needs to be taken into consideration and it can depend on the material brittleness, hardness, toughness and strength [54]. The angle limit value can even decrease to 10° for the most wear resistant materials cut by the standard abrasive materials (garnet, olivine). More intense studies of this problem are proceeding now and will be presented in future. The usual results for equipment and settings used in the Laboratory of Liquid Jet at the VSB—Technical University of Ostrava (see next sections) can be summarized in this way: the thickness limit for the declination angle 45° is about 30 mm, for the angle 22.5° it is 60 mm and for the angle 15° it is approximately 120 mm. Over the thickness of 120 mm it is necessary to use traverse speed producing the declination angle below 10°, otherwise the cutting process is disrupted and the jet rebounds from the kerf bottom.

The model was used for determination of the product deformation presented, namely, in [55] and the influence of the taper presented in [19]. The diameter of the cylindrical sample can then be calculated from this equation:(26)$$D=2\left[\sqrt{{\left(\frac{2}{5}H\tan\vartheta \right)}^{2}+R^{2}}+\frac{2}{5}H\tan\varphi \right]-d_{a}$$

The main benefit of the presented theoretical base is quite simple and relatively precise transformation of knowledge from the known and proven stages into new ones. Such transformations were presented during precision investigations in [62,63,64]. One of the most important is the transformation of the traverse speed limit for the different pressure and the abrasive mass flow rate. The subsequent equation can be used for this operation:(27)$$v_{Plim2}=\frac{\sqrt{\rho_{j2}p_{o2}^{3}}}{\sqrt{\rho_{j1}p_{o1}^{3}}}v_{Plim1}$$

Calculations of the traverse speed limits for the different jet parameters are very useful for comparison of results among workplaces with different experimental (manufacturing) facilities.

## 5. Simplification of the Model for Implementation to the Control Systems

In spite of the fact that the above presented model is derived by applying physical and mathematical procedures describing the objective phenomena and processes, its application in practice may appear too complicated and demanding. Therefore, the reduced form for rapid application is presented in this part of the article. The simplification is based on the fact that a very limited number of all the parameters, factors and characteristics influencing the machining process are actually changed in the AWJ applications in practice.

First of all, the important statement needs to be noted. The operation parameters like the nozzle/orifice diameter, the mixing chamber configuration, the focusing tube diameter, used liquid (predominantly water), used abrasive including its sizing as well as the operating pressure and the stand-off distance are very often constant for any applications in the given workplace in practice. Therefore, the simplified forms of the equations can be prepared by implementing the preset parameters in their numerical expressions. The very often used combination of these parameters in our laboratory/workplace is presented here as an example:
Operating pressure380 MPaStand-off distance2 mmNozzle (orifice) diameter0.25 mmFocusing tube diameter1.02 mmFocusing tube length76.2 mmUsed liquidwaterUsed abrasiveAustralian garnetAbrasive sizing80 mesh (0.25 mm) *Abrasive mass flow rate0.25 kg·min^−1^* Comment: *Average grain size of Australian garnet 80 mesh has been measured in laboratories at the VSB—Technical University of Ostrava several times on different measuring devices for particle size anal-yses. The average value 0.25 mm has been determined and, therefore, it is used now in our calculations, although some conversion tables present lower values (often below 0.2 mm)*.

Using these data and the usual geometry of the water nozzle, the mixing chamber and the focusing tube (Paser II), the subsequent fixed values, summarized in Table 1, can be determined (e.g., in Excel).

Examples of calculation of both the depth limit of penetration and the traverse speed limit are presented for selected metal materials (high strength steel, tool steel, stainless steel, very abrasive resistant Hardox 500 steel, copper, brass and duralumin) and rock materials (hard sandstone, limestone, marble, granite and strong granite). Mild sandstone was not used for these experiments, because it is so easily disintegrated that it can be cut very efficiently even by an almost pure water jet. Respective material characteristics are summarized in Table 2 and Table 3.

As can be seen, the higher the value of the “response” to the AWJ the more difficult the cutting. However, it is also evident that the “response” factor is also dependent on the thickness of material used for the testing cut. Therefore, such a modification of the theoretical model was sought, which will allow excluding of the influence of the absolute size of the “response” of the AWJ. Recent results show that it is not necessary to apply the whole presented theoretical model. It can be accepted as the proven theoretical background and the practical applications can be based on the most recent knowledge expressed through Equations (21)–(25) and (27) or others derived on the base of similarity. Demonstration of this proposition is a content of the subsequent section.

## 6. Comparison of Model Results with Experiments and Discussion

Experimental works were performed in the Laboratory of Liquid Jet at the VSB—Technical University of Ostrava. The equipment consisted of the commercial x-y table with manually handled z-axis PTV WJ 1020-1Z-EKO (PTV s.r.o., Hostivice, Czech Republic) and the Flow X5 pump (Flow Int., Seattle, WA, USA). Special research devices for studying of the tilted jets and turning were added over the years. However, none of them allows a controlled change of the jet impact angle on the material and stand-off distance from the material. The pump maximum flow rate of 1.9 L/min does not allow using orifices with diameters greater than 0.25 mm.

Practical applications can be demonstrated in two main ways. The first one is based on a comparison of the measured sample deformation (caused by the trailback and the taper) with the calculated one. This comparison was presented in publications aimed at the deformation of samples prepared by AWJ machining, namely, [63,64]. Some new results demonstrating the calculation results are summarized in Table 4. Tilting is just compensating of the trailback, because the experimental equipment in the Laboratory of Liquid Jet at the VSB—Technical University of Ostrava is prepared for cutting of column samples, this enables tilting in only the tangential plane to the cylindrical sample shell, not in the radial direction. However, it can be assumed that the influence of the trailback is compensated for, and, therefore, the input diameter $D_{it}$ is equal to the output one without influence of the “taper”. The total theoretical diameter is then the sum of the theoretical input diameter $D_{it}$ and the difference caused by the “taper” T (both these values were calculated from the theoretical model summarizing the deformation of the sample—Equation (26)). Because the combined uncertainty of the sample measurements is 1.6%, it is evident that the deviation of the respective experimental and theoretical values lies in this interval.

The second demonstration is based on the comparison presented in Table 5. The respective experiments were performed with the AWJ facility installed in the Laboratory of Liquid Jet at the VSB—Technical University of Ostrava.

All presented calculations are based on transformations of the traverse speed limits from the proven measured ones into the new states determined for the different conditions only through calculations from the presented model. The basic traverse speed limit $v_{PlimB}$ is determined for the settings presented in the beginning of Section 5 (the classical settings). The samples demonstrating the strength of the theoretical model were cut with just one change in these settings: instead of the focusing tube with diameter $d_{a}=1.02\;\mathrm{mm}$ the one with diameter $d_{aC}=0.76\;\mathrm{mm}$ was used. The changed traverse speed limit $v_{PlimN}$ was then determined. The new traverse speed limit was calculated from this equation, prepared from the theoretical model:(28)$$v_{PlimN}=v_{PlimB}\frac{d_{a}}{d_{aC}}\frac{a_{nC}}{a_{n}},$$
where $v_{PlimB}$ is the traverse speed limit for the basic set of experimental conditions and $v_{PlimN}$ is the traverse speed limit after changing the focusing tube; *d_a_* and *d_aC_* are the respective diameters of the original and the changed focusing tubes; *a_n_* and *a_nC_* are the medium sizes of abrasive particles after mixing process for the original and the changed focusing tubes, respectively (data based on the studies published in [36,65]—$a_{n}=24.95\;\mu m$, $a_{nC}=22.15\;\mu m$).

Examples of the mean declination angles are presented on the photos of the experimental samples in Figure 1 and Figure 2. These selected metal samples are ones from the series of three cuts performed under identical conditions. Respective values measured for all three samples of selected metals are summarized in Table 6.

It can be seen that both the relative difference between the theoretical and the experimental values is below 5% (Table 5) and the relative uncertainty of measurement on respective samples is also below 5% (Table 6). Therefore, the agreement of the theoretical and the respective experimental results can be considered as very good.

The presented model of the material cutting by AWJ is highly applicable on homogeneous and quasi-homogeneous materials—metals, rocks, concretes (if not reinforced or containing materials with extremely different mechanical properties), ceramics, glass and homogeneous plastics. The model needs additional “modification” for reinforced, sandwich, honeycomb or other non-homogeneous structures; it is applicable for these structures when they are well described and both the position and influence of the inhomogeneity is very predictable. The appropriate changes of the jet size when moving from one material to another need to be calculated and the respective power loss needs to be evaluated and taken into account.

The presented results indicate that use of the partial relations derived from the model (theory) can be very effective in prediction of either the results of machining with certain settings or calculation of the appropriate settings for achieving the required machining results. Therefore, these partial equations, which can be derived from the basic theoretical description and the respective model, can be used for preparation of control software with very high precision.

## 7. Conclusions

The presented theoretical model includes almost all parameters of either the plain water jet or the abrasive water jet, the machined material and other factors influencing the quantity and the quality of the workpiece. The resulting equations yield many opportunities to use only the partial relations among few parameters or factors for a transfer of proven knowledge to the changed conditions. Some of these examples are presented and the difference between the respective theoretical and experimental values is below 5%. Partial relations can be used in practice for preparation of control programs that can calculate very exact settings of changeable parameters from the proven results obtained with other settings or some “default settings”.

## Figures and Tables

**Figure 1 materials-14-04032-f001:**
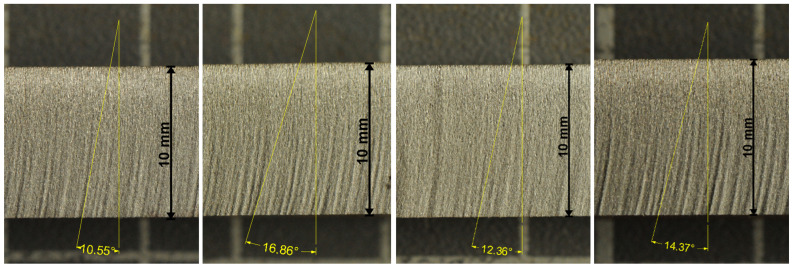
Steels and the respective average angles—the order from the left to the right (WRN norm): high strength steel (1.7131), tool steel (1.2436), stainless steel (1.4541), Hardox 500 (trademark of the SSAB).

**Figure 2 materials-14-04032-f002:**
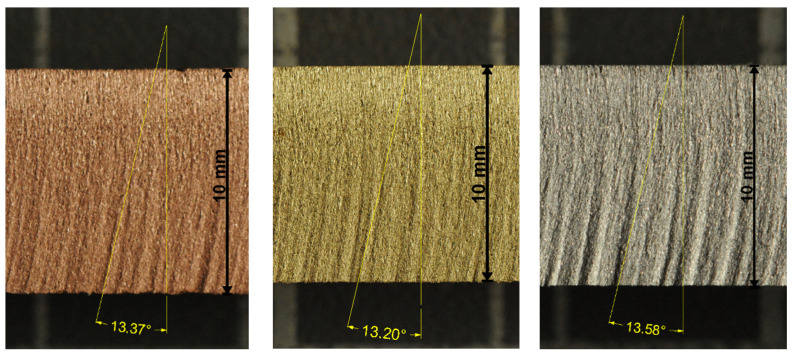
Non-ferrous metals and the respective average angles—the order from the left to the right (WRN norm): copper (2.0060), brass (2.0402), duralumin (3.1325).

**Table 1 materials-14-04032-t001:** Variables determined for the tested configuration of the injection abrasive water jet.

Variable	Value	Unit	Variable	Value	Unit	Variable	Value	Unit
*μ_o_*	0.5981	-	*C_A_*	0.0033	-	*c_o_*	1484	m·s^−1^
*γ_o_*	0.8993	-	*S_P_*	1.0000	-	*c*	4600	m·s^−1^
*C_D_*	0.2000	-	*f*	0.1000	-	*v_o_*	640	m·s^−1^
*C_1_*	0.7926	-	*l_a_*	0.0762	m	*v_a_*	378	m·s^−1^
*C_2_*	1.0000	-	*a_o_*	0.2500	mm	*ρ_j_*	1095	kg·m^−2^
*C_3_*	0.9238	-	*a_n_*	24.951	μm	*p_j_*	78.19	MPa
*C_4_*	0.9136	-	*E_P_*	2.8350	J·m^−2^	*ξ_j_*	1.142	m^−1^

**Table 2 materials-14-04032-t002:** Parameters of metals; all samples were 10 mm thick.

Material (WRN/DIN Norm)	Yield Strength MPa	Density kg·m^−3^	Response to AWJ for the Water Nozzle Diameter	Response to AWJ for the Focusing Tube Diameter
High strength steel (1.7131/16 MnCr 5)	880	7746	120	39
Tool steel (1.2436/X210 CrW 12)	656	7674	108	36
Stainless steel (1.4541/X6 CrNiTi 18 10)	515	7521	100	34
Hardox 500 trademark of the SSAB	1679	7524	171	50
Copper (2.0060/E-Cu57)	211	8687	67	29
Brass (2.0402/CuZn40Pb2)	393	8364	73	30
Duralumin (3.1325/AlCu4MgSi)	419	2784	45	15

**Table 3 materials-14-04032-t003:** Parameters of rocks; all samples were 30 mm thick.

Material	Compressive Strength MPa	Grain Size μm	Density kg·m^−3^	Response to AWJ for the Nozzle Diameter	Response to AWJ for the Focusing Tube Diameter
Sandstone	150	0.52	2590	183	80
Limestone	85	0.51	2420	156	70
Marble	100	0.52	2650	165	75
Granite	188	0.69	2557	201	83
Strong granite	291	0.40	3041	376	129

**Table 4 materials-14-04032-t004:** Comparison of the theoretical calculations and the respective experimental data for the tilted cutting head.

Material	$v_{Ptilt}$ mm/min	*D_ie_* mm	*D_oe_* mm	*D_it_* mm	*T* mm	*D_ot_* mm	Relative Difference *D_i_*; *D_o_*
High strength steel	128	9.33	9.72	9.29	0.36	9.65	0.43%; 0.72%
Tool steel	93	9.34	9.66	9.31	0.36	9.67	0.32%; 0.10%
Stainless steel	116	9.32	9.63	9.28	0.37	9.65	0.43%; 0.21%
Hardox 500	105	9.32	9.62	9.33	0.35	9.68	0.11%; 0.62%
Copper	221	9.30	9.72	9.29	0.37	9.66	0.11%; 0.62%
Brass	219	9.31	9.71	9.30	0.36	9.66	0.11%; 0.52%
Duralumin	443	9.30	9.69	9.28	0.37	9.65	0.22%; 0.41%

**Table 5 materials-14-04032-t005:** Comparison of the theoretical calculations and the respective experimental data.

Material	$v_{PlimB}$ mm/min	$v_{PlimN}$ mm/min	Theoretical Angle (°)	Experimental Angle (°)	Relative Difference
High strength steel	220	262	10.60	10.55 ± 0.37	0.5%
Tool steel	160	191	17.10	16.86 ± 0.54	1.4%
Stainless steel	200	238	12.23	12.36 ± 0.25	1.0%
Hardox 500	180	214	14.33	14.37 ± 0.15	0.3%
Copper	380	453	13.21	13.37 ± 0.17	1.2%
Brass	376	448	13.42	13.20 ± 0.03	1.7%
Duralumin	760	906	13.21	13.58 ± 0.11	2.7%

**Table 6 materials-14-04032-t006:** The declination angles measured for experimental samples in degrees (10 measurements for each case).

Material	Experiment I.	Experiment II.	Experiment III.	Average Value	Absolute Uncertainty	Relative Uncertainty
High strength steel	10.62	10.97	10.06	10.55	±0.37	3.6%
Tool steel	16.86	16.20	17.53	16.86	±0.54	3.2%
Stainless steel	12.25	12.12	12.71	12.36	±0.25	2.0%
Hardox 500	14.22	14.32	14.59	14.37	±0.15	1.1%
Copper	13.32	13.60	13.18	13.37	±0.17	1.3%
Brass	13.21	13.22	13.16	13.20	±0.03	0.2%
Duralumin	13.46	13.60	13.56	13.58	±0.11	0.8%

## Data Availability

Data sharing is not applicable to this article.

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
