# Peer review of "Revised Model of Abrasive Water Jet Cutting for Industrial Use"

_materials, 2021, doi:10.3390/ma14144032_

Round 1

Reviewer 1 Report

This paper presents a revised model of abrasive water jet cutting in the industry application. The author’s work shows the improved model has a potential tool for the preparation of the control software with high precision. The paper presents enough background introduction and the current improvement. Detailed reasoning and results analysis were presented. The explanation was in a logical order and contained a few grammar errors. I think this work will be beneficial to the community in the AWJ machining, as it provides a tool to control the speed. A few comments have been listed below.

On page 10, I would suggest putting the parameters in the table format.

Figure 1, it would be better to specify the type of the tool steel, stainless steel, i.e., AISI 304 stainless steel.

Figure 1, I think it is necessary to put the scale bar at the bottom of the figure to have an idea of the metals.

Similar to the Figure 2.

Reviewer 2 Report

Dear Author,

Thank you for getting to know your work.
The manuscript is very interesting and deals with a very important and current problem of the waterjet cutting process. The work has an original character, the aim was to adjust the model and verify its fit in experimental tests. Overall, the job is good and has high scientific and application value. Nevertheless, I found a few points that please clarify and complete in the text:

In chapter 3 (lines 210-220) the author writes about various materials, what materials does the presented model take into account?

Lines 342-355, the author lists different values ​​of the limit angles, are they theoretical or experimentally determined? If they come from literature, it should be referred to.
The author writes that the materials are thick, but this is a very general statement, please give the range of thickness, some values.

In chapter 5, the author specifies the parameters of the machine. There is no information about what the machine is, whether it is a mass-produced model or a research device (prototype), etc.
It is worth mentioning because of the possibility of repeating the research, mapping by other researchers to recreate the tests and possible verification of the obtained results.
The results of the model mapping, according to the author, are very promising (lines 470-473).

Thank you.
